# *Aspergillus fumigatus* transcription factor ZfpA regulates hyphal development and alters susceptibility to antifungals and neutrophil killing during infection

Taylor J. Schoen[1,2], Dante G. Calise[1,3], Jin Woo Bok[1], Morgan A. Giese[1,4], Chibueze D. Nwagwu[5], Robert Zarnowski[1,6], David Andes[1,6], Anna Huttenlocher[1,7]*, Nancy P. Keller[1,8]*

1 Department of Medical Microbiology and Immunology, University of Wisconsin-Madison, Madison, Wisconsin, United States of America, 2 Comparative Biomedical Sciences Graduate Program, University of Wisconsin-Madison, Madison, Wisconsin, United States of America, 3 Microbiology Doctoral Training Program, University of Wisconsin-Madison, Madison, Wisconsin, United States of America, 4 Cellular and Molecular Biology Graduate Program, University of Wisconsin-Madison, Madison, Wisconsin, United States of America, 5 Emory University School of Medicine, Atlanta, Georgia, United States of America, 6 Department of Medicine, University of Wisconsin-Madison, Madison, Wisconsin, United States of America, 7 Department of Pediatrics, University of Wisconsin-Madison, Madison, Wisconsin, United States of America, 8 Department of Plant Pathology, University of Wisconsin-Madison, Madison, Wisconsin, United States of America

* huttenlocher@wisc.edu (AH); npkeller@wisc.edu (NPK)

## Abstract

Hyphal growth is essential for host colonization during *Aspergillus* infection. The transcription factor ZfpA regulates *A. fumigatus* hyphal development including branching, septation, and cell wall composition. However, how ZfpA affects fungal growth and susceptibility to host immunity during infection has not been investigated. Here, we use the larval zebrafish-*Aspergillus* infection model and primary human neutrophils to probe how ZfpA affects *A. fumigatus* pathogenesis and response to antifungal drugs *in vivo*. ZfpA deletion promotes fungal clearance and attenuates virulence in wild-type hosts and this virulence defect is abrogated in neutrophil-deficient zebrafish. ZfpA deletion also increases susceptibility to human neutrophils *ex vivo* while overexpression impairs fungal killing. Overexpression of ZfpA confers protection against the antifungal caspofungin by increasing chitin synthesis during hyphal development, while ZfpA deletion reduces cell wall chitin and increases caspofungin susceptibility in neutrophil-deficient zebrafish. These findings suggest a protective role for ZfpA activity in resistance to the innate immune response and antifungal treatment during *A. fumigatus* infection.

## Author summary

*Aspergillus fumigatus* is a common environmental fungus that can infect immunocompromised people and cause a life-threatening disease called invasive aspergillosis. An

**Data Availability Statement:** All relevant data is within the manuscript and supporting information files.

**Funding:** This work was supported by R35GM118027-01 from the National Institute of General Medical Sciences (NIGMS) of the National Institutes of Health (NIH) to A.H. and 5 R01 AI150669-03 from the National Institute of Allergy and Infectious Diseases (NIAID) of the NIH to N.P. K. T.J.S. was supported by the National Institute on Aging of the National Institutes of Health under Award Number T32AG000213. The content is solely the responsibility of the authors and does not necessarily represent the official views of the NIH. The funders had no role in study design, data collection and analysis, decision to publish, or preparation of the manuscript.

**Competing interests:** The authors have declared that no competing interests exist.

important step during infection is the development of *A. fumigatus* filaments known as hyphae. *A. fumigatus* uses hyphae to acquire nutrients and invade host tissues, leading to tissue damage and disseminated infection. In this study we report that a regulator of gene transcription in *A. fumigatus* called ZfpA is important for hyphal growth during infection. We find that ZfpA activity protects the fungus from being killed by innate immune cells and decreases the efficacy of antifungal drugs during infection by regulating construction of the cell wall, an important protective layer for fungal pathogens. Our study introduces ZfpA as an important genetic regulator of stress tolerance during infection that protects *A. fumigatus* from the host immune response and antifungal drugs.

## Introduction

*Aspergillus spp*. are common environmental fungi that are not considered a significant risk to healthy individuals. However, inhalation of airborne *Aspergillus* spores can lead to invasive and disseminated hyphal growth, damaging inflammation, and death in immunocompromised populations, where mortality rates exceed 50% [1]. *Aspergillus fumigatus* is the most frequent cause of invasive aspergillosis (IA), one of the most important fungal infections of humans [2].

Anti-*Aspergillus* immunity is largely mediated by innate immune cells, which are sufficient to prevent formation of the tissue invasive hyphae characteristic of IA [3]. In an immunocompetent host, spore germination and subsequent hyphal growth are limited by macrophages and neutrophils, respectively. Hyphae stimulate potent fungicidal functions in neutrophils such as reactive oxygen species, extracellular trap formation, and cytotoxic granule release, resulting in fungal clearance [3,4]. In addition to being targeted by neutrophils, the hyphal growth stage is the primary target of antifungal drugs used to treat IA [5,6]. Left unchecked, *A. fumigatus* can form branching hyphal networks within host tissues. *In vitro*, hyphal branching is regulated by endogenous signaling pathways [7–11], like those involved in cell wall construction [12,13], and environmental cues such as physical contacts with neutrophils [14]. However, the mechanisms that regulate hyphal morphology and growth during infection are not fully understood.

In a recent screen for transcriptional regulators of hyphal branching, we identified the C2H2 zinc finger transcription factor ZfpA (AFUB_082490) [11]. ZfpA overexpression induces hyperbranched hyphae with increased septation and cell wall chitin. Conversely, ZfpA null hyphae have reduced branching, septation, and cell wall chitin. In addition to its role in hyphal development, ZfpA has been implicated in *A. fumigatus* response to stressors such as high calcium and the antifungals voriconazole and caspofungin [15–18]. However, whether ZfpA affects fungal growth and susceptibility to host immunity and antifungal treatment during infection remains unclear.

The larval zebrafish is an ideal model to directly observe how ZfpA affects fungal development and immune cell behavior. The innate immune response of zebrafish larvae to *Aspergillus* shares many similarities with that of mammals, including fungal clearance by macrophages and neutrophils, with the unique advantage of repeated live-imaging of larvae over the course of multi-day infections [19–24]. Additionally, *A. fumigatus* infections in zebrafish can be successfully treated with clinically relevant antifungals [25], allowing us to screen for changes in drug susceptibility of ZfpA mutants *in vivo*.

Here, we sought to determine whether ZfpA-mediated changes to hyphae could impact tissue invasion, resistance to host defenses, and antifungal susceptibility during infection. Using a combination of *in vivo* zebrafish experiments and human neutrophil killing assays, we show

that loss of ZfpA does not impede tissue invasion during infection but limits virulence by increasing fungal susceptibility to neutrophil killing. Further, ZfpA deletion increases susceptibility to caspofungin, but not voriconazole, during infection of neutrophil-deficient hosts. Notably, ZfpA overexpression decreases susceptibility to both neutrophil killing and antifungal treatment. We found that ZfpA confers protection against caspofungin via regulation of chitin synthesis during hyphal development, offering mechanistic insight into the function of this transcription factor during echinocandin exposure. Together, these findings establish a role for ZfpA in tolerance to stress induced by the host immune response and antifungal drugs during infection.

## Results

### ZfpA regulates virulence and fungal burden but does not affect immune cell recruitment in wild-type hosts

We hypothesized that the effects of ZfpA on hyphal development may impact tissue invasion and resistance to host defenses during infection. Among other phagocyte defects, neutrophil-deficiency or neutropenia is a major risk factor for IA development [26]. Therefore, to test whether ZfpA is important for pathogenesis in a clinically relevant host background, we used a larval zebrafish model of a human leukocyte adhesion deficiency in which neutrophils express a dominant-negative Rac2D57N mutation (*mpx:rac2D57N*) that impairs recruitment to infection and host survival [19,27]. We found that virulence of Δ*zfpA* was attenuated in wild-type control larvae (*mpx:rac2wt*) while virulence of OE::*zfpA* was similar to WT CEA10 (Fig 1). However, in neutrophil-defective Rac2D57N larvae, Δ*zfpA* had similar virulence compared to WT CEA10 but remained less virulent than OE::*zfpA* (Fig 1). We observed similar virulence trends of ZfpA mutants in the less virulent Af293 background (S1 Fig).

To better understand the virulence defect of Δ*zfpA* in wild-type larvae and to determine the impact of ZfpA on fungal growth and inflammation during infection, we injected RFP-expressing WT CEA10, Δ*zfpA*, or OE::*zfpA* strains into immunocompetent, transgenic zebrafish larvae with fluorescent neutrophils and macrophages (Tg(*lyz:BFP/ mpeg1:EGFP*)). We then imaged the infected hindbrain ventricle at 24, 48, 72, and 96 hours post infection (hpi) to track fungal burden and phagocyte recruitment. ZfpA had no effect on spore germination rate, with all strains germinating by 48 hpi (Fig 2A and 2B). The fungal burden of each strain was similar up to 72 hpi, but at 96 hpi larvae infected with Δ*zfpA* had significantly reduced fungal burden relative to WT CEA10 and OE::*zfpA* (Fig 2A and 2C). As there were no significant differences in the scale of neutrophil or macrophage recruitment over the course of infection (Fig 2A, 2D and 2E), we suspected that the reduction in fungal burden and virulence could be due to increased susceptibility of Δ*zfpA* to leukocyte-mediated killing.

### ZfpA regulates resistance to human neutrophils

While both macrophages and neutrophils respond to *A. fumigatus* infection, neutrophils are the primary immune cell responsible for clearing hyphal growth [19,28]. To directly test whether ZfpA alters resistance to neutrophil-mediated killing, we isolated primary human neutrophils for co-incubation with WT CEA10, Δ*zfpA*, and OE::*zfpA* germlings and imaged neutrophil-hyphal interactions every 3 min for 12 h. In these movies we were able to measure hyphal death through loss of cytoplasmic RFP signal and observe antimicrobial neutrophil behaviors such as directed migration, phagocytosis, swarming, and the release of cellular contents (Fig 3A, S1–S4 Movies). We saw that Δ*zfpA* hyphae were remarkably susceptible to neutrophil attack as most germlings died within 1 h and none survived longer than 5 h of co-

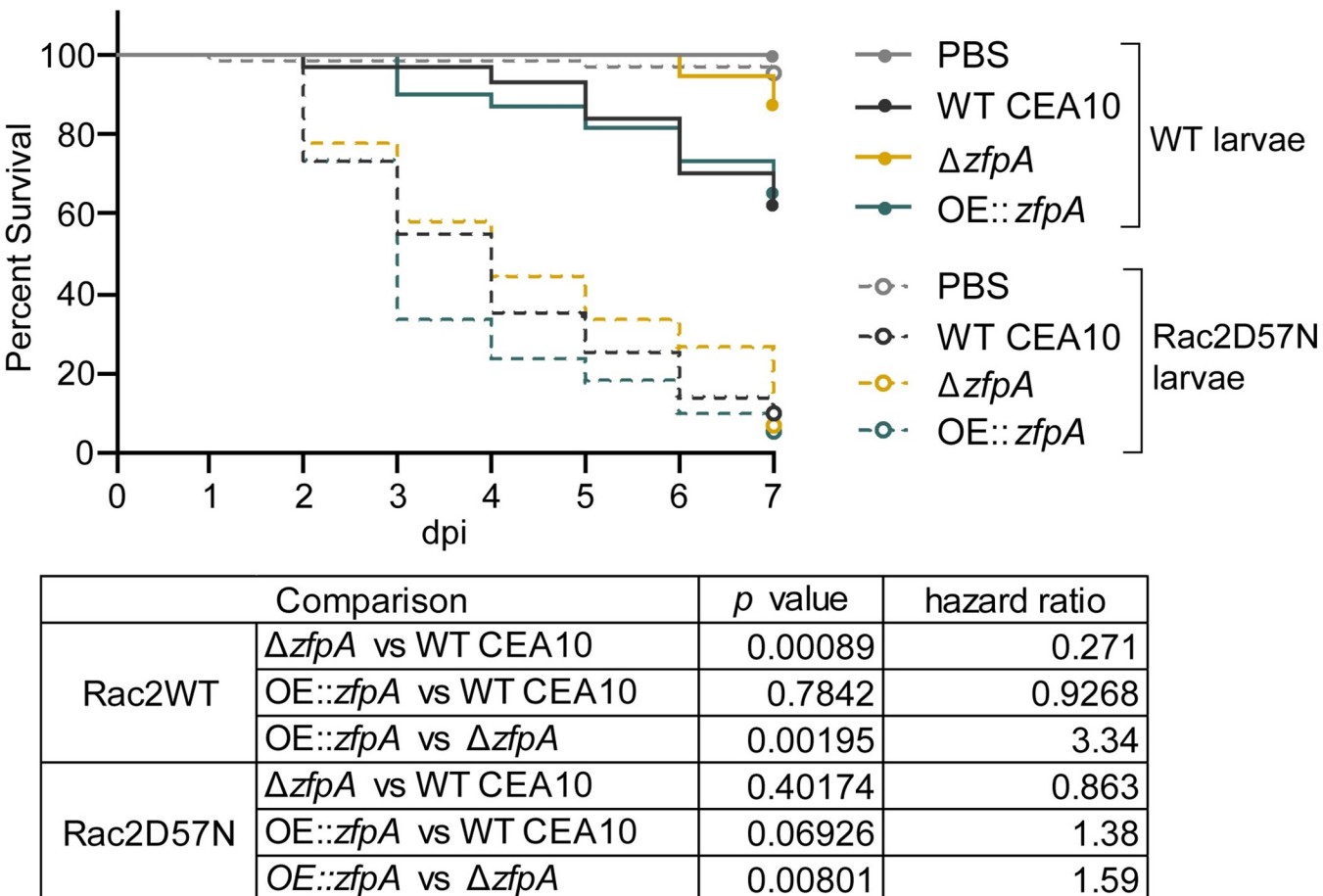

**Fig 1. ZfpA impacts virulence in wild-type but not neutrophil-deficient hosts.** Survival analysis of larvae with the dominant negative Rac2D57N neutrophil mutation (neutrophil-deficient) or wild-type siblings injected with PBS, WT CEA10, Δ*zfpA*, or OE::*zfpA* strains. WT larvae average spore dose injected: WT CEA10 = 60, Δ*zfpA* = 50, OE::*zfpA* = 62. Rac2D57N larvae average spore dose injected: WT CEA10 = 52, Δ*zfpA* = 54, OE::*zfpA* = 53. Results represent pooled data from 3 independent replicates. n = 71–72 larvae per condition. *p* values and hazard ratios calculated by Cox proportional hazard regression analysis.

incubation (Fig 3B, S2 Movie). Conversely, OE::*zfpA* hyphae (S3 Movie) were best able to withstand neutrophil attack and maintained cytoplasmic RFP signal longer than Δ*zfpA* (S2 Movie) and WT CEA10 (S1 Movie, Fig 3B). We observed no differences between strains in the timing of neutrophil contacts, suggesting that the initial migratory response of neutrophils to germlings is not impacted by ZfpA (Fig 3C). As the ZfpA mutants have aberrant branching patterns that could alter their ability to evade neutrophils, we measured the percent of germlings able to escape neutrophil contact by extending hyphae outside of surrounding neutrophil clusters. We found that Δ*zfpA* germlings never escaped surrounding neutrophils, while WT CEA10 and OE::*zfpA* escaped at similar frequencies, suggesting that the enhanced ability of OE::*zfpA* to withstand neutrophil activity is not due to an increased ability to evade neutrophils via branch production (Fig 3D). These data also suggest that susceptibility to neutrophil killing underlies the virulence defect of Δ*zfpA* in wild-type hosts and that ZfpA confers protection from host defenses.

It is possible that Δ*zfpA* and OE::*zfpA* elicit distinct neutrophil effector mechanisms that contribute to differential killing of these strains. Formation of neutrophil extracellular traps (NETs) is a well-documented response to *A. fumigatus* hyphae that contributes to inflammation and control of fungal growth [29–31]. To test whether Δ*zfpA* and OE::*zfpA* trigger

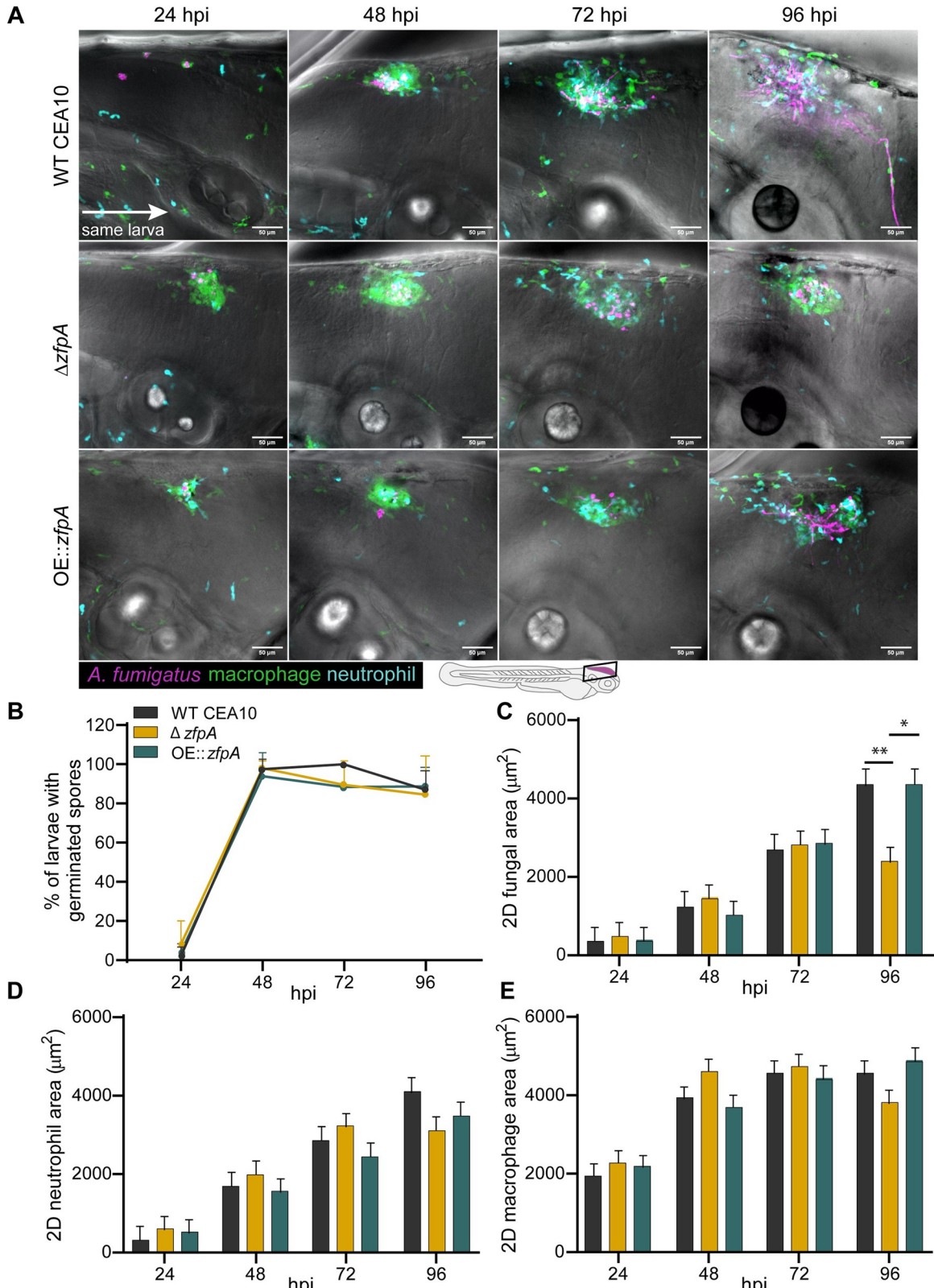

**Fig 2. ZfpA controls fungal burden but does not affect immune cell recruitment in wild-type hosts.** 2-day post fertilization wild-type larvae with fluorescent macrophages (GFP) and neutrophils (BFP) were infected with RFP-expressing WT CEA10, Δ*zfpA*, or OE::*zfpA*

strains. Larvae were imaged with confocal microscopy at 24, 48, 72, and 96 hours post infection (hpi). (A) Representative images of fungal growth and immune cell recruitment in the same larva at 24–96 hpi. Images represent maximum intensity projections of z-stacks. Scale bar = 50 μm. (B) Mean percentage of larvae with germinated spores at 24–96 hpi. Dots and error bars represent mean+s.d. (C) 2D fungal (RFP) area at 24–96 hpi. (D) 2D neutrophil (BFP) area 24–96 hpi. (E) 2D macrophage (GFP) area at 24–96 hpi. Bars in (C-E) represent lsmeans+s.e.m. Results represent data pooled from 4 independent experiments. n = 45–48 larvae per condition. *p* values calculated by ANOVA with Tukey's multiple comparisons. *\*p<0.05, \*\*p<0.01.*

different levels of NETosis by human neutrophils, we co-incubated germlings and neutrophils for 6 h and measured extracellular DNA content using SYTOX Green staining. We found that neutrophils produced similar amounts of extracellular DNA in response to WT CEA10, Δ*zfpA*, and OE::*zfpA* (Fig 4A), suggesting that differences in survival of Δ*zfpA* and OE::*zfpA* germlings is not due to increased or decreased NET production, respectively.

We next assessed whether Δ*zfpA* and OE::*zfpA* alter neutrophil inflammatory signaling and degranulation. The pro-inflammatory cytokine IL-8 (CXCL8) is a potent neutrophil chemoattractant while myeloperoxidase (MPO) is a primary component of neutrophil granules that has both cytotoxic and immunomodulatory functions [32,33]. We measured neutrophil release of IL-8 and MPO following 3 h of co-incubation with germlings of WT CEA10, Δ*zfpA*, and OE::*zfpA*. Neutrophils released IL-8 and MPO in response to all strains, however, levels of IL-8 and MPO were similar among all conditions (Fig 4B and 4C). Collectively, our results suggest that neutrophil activation in response to *A. fumigatus* is not affected by ZfpA.

Production of reactive oxygen species (ROS) has both fungicidal and immunomodulatory functions during *Aspergillus* infection and is important for efficient pathogen clearance. To

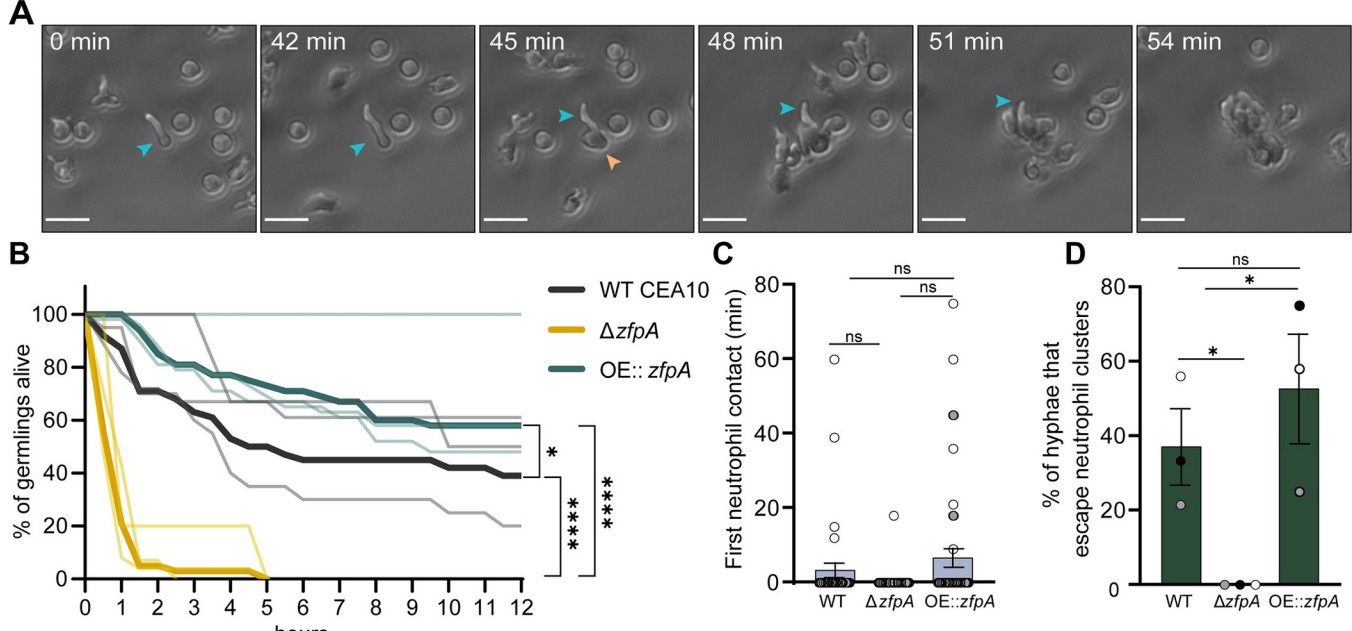

**Fig 3. ZfpA promotes resistance to neutrophil killing.** Outcomes of human neutrophil interactions with WT CEA10, Δ*zfpA*, and OE::*zfpA* following 12 h of co-incubation. Neutrophils were added to *A. fumigatus* germlings (neutrophil:spore 100:1) in 24-well plates and images were acquired every 3 min for 12 h. (A) Representative images of neutrophils detecting and tightly clustering around an *A. fumigatus* (OE::*zfpA*) germling within the first hour of co-incubation. Blue arrow indicates visible germling. Orange arrow indicates first neutrophil contact. Scale bar = 20 μm. (B) Percent of germlings alive determined by presence of cytoplasmic RFP signal at 30 min intervals over 12 h. Thin lines represent data from 3 independent experiments, thick lines represent pooled data. n = 38–47 germlings per strain. *p* values calculated by Cox proportional hazard regression analysis. (C) Time of first neutrophil contact with germling. Dots represent individual germlings color coded by replicate and bars represent mean±s.e.m. Statistical comparisons calculated by *t*-tests. (D) Percent of germlings able to "escape" neutrophil contact by extending hyphae outside of surrounding neutrophil clusters. Bars represent mean ±s.e.m. Dots represent independent experiments. *p* values calculated by *t*-tests. *\*p<0.05, \*\*\*\*p<0.0001.*

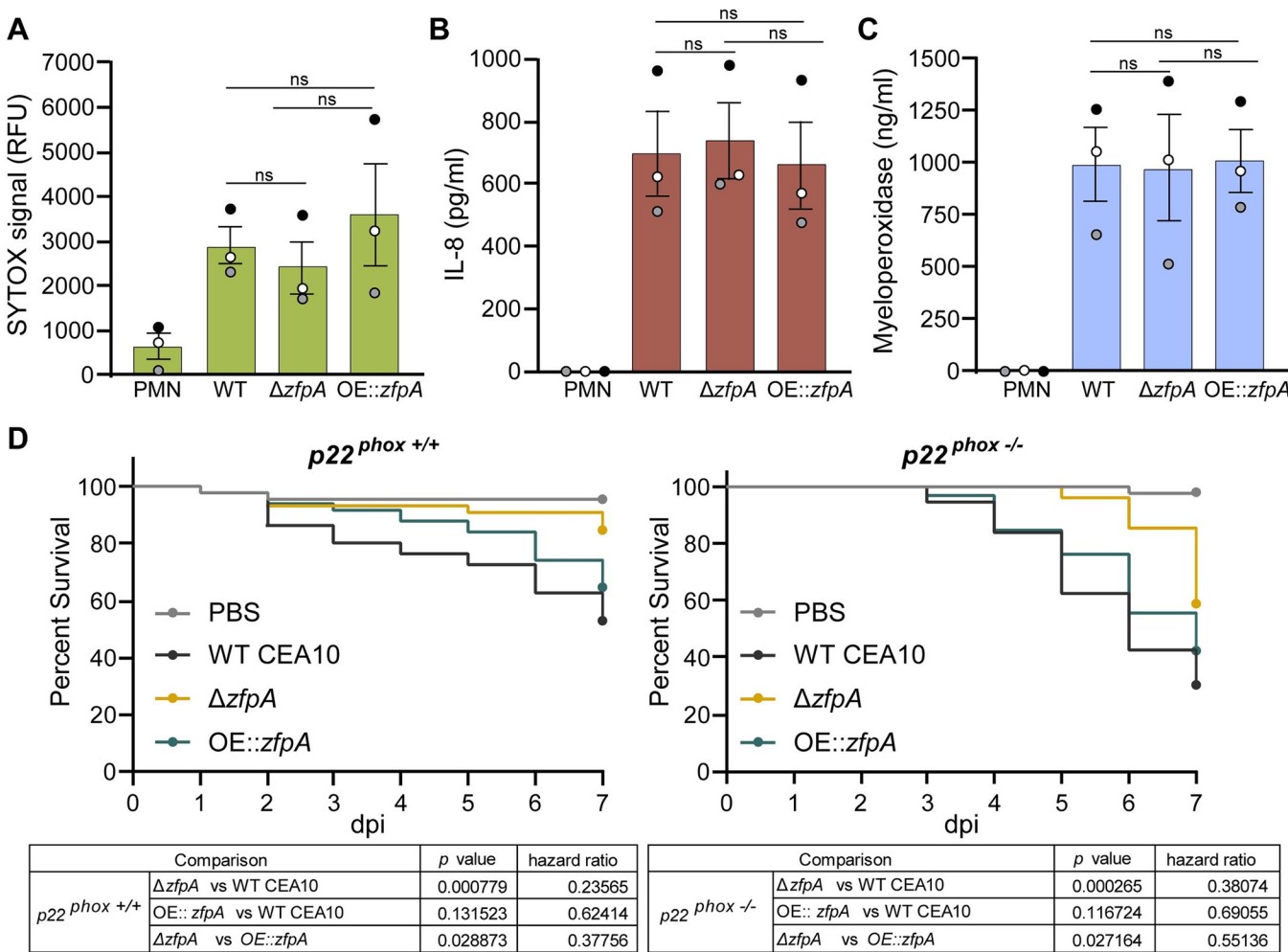

**Fig 4. ZfpA has minimal impact on neutrophil effector functions or fungal susceptibility to phagocytic ROS.** (A) Neutrophil extracellular trap formation in response to WT CEA10, Δ*zfpA*, and OE::*zfpA* hyphae after 6 h of co-incubation (neutrophil:spore 2:1). Bars represent mean±s.e.m. of relative fluorescent units (RFU) of Sytox Green signal from 3 independent experiments (dots). (B) ELISA quantification of IL-8 production by neutrophils in response to WT CEA10, Δ*zfpA*, and OE::*zfpA* hyphae after 3 h of co-incubation (neutrophil:spore 2:1). Bars represent mean±s.e.m. from 3 independent experiments (dots). (C) ELISA quantification of myeloperoxidase release by neutrophils in response to WT CEA10, Δ*zfpA*, and OE::*zfpA* hyphae after 3 h of co-incubation (neutrophil:spore 2:1). Bars represent mean±s.e.m. from 3 independent experiments (dots). Statistical comparisons calculated by *t*-tests. (D) Survival analysis of PHOX-deficient larvae or wild-type siblings injected with PBS, WT CEA10, Δ*zfpA*, or OE::*zfpA* strains. *p22*^+/+ larvae average spore dose injected: WT CEA10 = 45, Δ*zfpA* = 40, OE::*zfpA* = 42. *p22*^-/- larvae average spore dose injected: WT CEA10 = 35, Δ*zfpA* = 41, OE::*zfpA* = 31. Results represent pooled data from 3 independent replicates. n = 45–59 larvae per condition. *p* values and hazard ratios calculated by Cox proportional hazard regression analysis.

test whether the enhanced susceptibility of Δ*zfpA* is dependent on neutrophil ROS, we utilized the *p22*^-/- larval zebrafish model which lacks ROS production in phagocytes [22]. Virulence of the Δ*zfpA* mutant was reduced relative to WT CEA10 and OE::*zfpA* in both *p22*^+/+ and *p22*^-/- larvae (Fig 4D), suggesting that ROS is not the primary mechanism responsible for enhanced killing of Δ*zfpA*.

## ZfpA contributes to cell wall integrity but is not implicated in osmotic or oxidative stress

Alterations in stress resistance in the ZfpA mutants could underpin differences in virulence and susceptibility to neutrophil-killing mechanisms. We therefore challenged ZfpA mutants

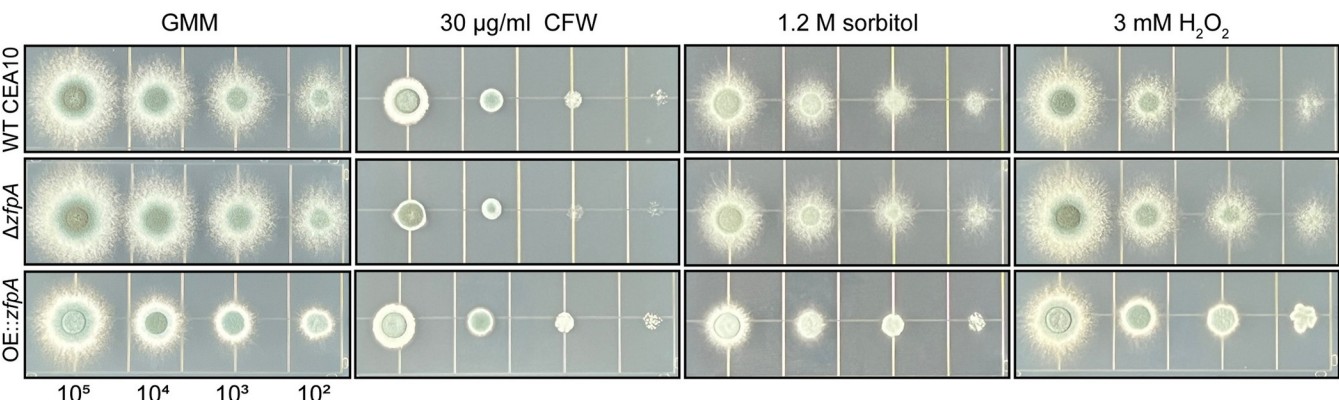

**Fig 5. ZfpA alters resistance to cell wall perturbation but not osmotic or oxidative stressors.** Spot-dilution assays to test susceptibility of ZfpA mutants to the cell wall stressor calcofluor white (CFW), osmotic stressor sorbitol, or the oxidative stressor $H_2O_2$. Spores were point-inoculated on solid glucose minimal medium (GMM) ± stressors at concentrations of $10^5 - 10^2$ and incubated for 48 hours at $37°C$. Images are representative of growth from 3 plates per condition.

with cell wall, osmotic, and oxidative stressors using spot-dilution assays. When grown on glucose minimal medium (GMM), colony morphology differs between WT CEA10, Δ*zfpA*, and OE::*zfpA*. Δ*zfpA* colonies have more filamentous growth around the colony edge while OE::*zfpA* colonies are more compact with less filamentous growth visible at the edges (Fig 5). ZfpA deletion was previously shown to increase susceptibility to the common cell wall stressor calcofluor white (CFW) which impairs cell wall integrity by disrupting assembly of chitin chains in the cell wall [11,34]. Accordingly, Δ*zfpA* showed increased susceptibility to CFW while OE::*zfpA* was more resistant (Fig 5). There were no clear differences between strains in susceptibility to the osmotic stressor sorbitol or the oxidative stressor $H_2O_2$ (Fig 5), suggesting cell wall defects as the primary driver of differential stress resistance in these mutants.

## ZfpA overexpression decreases voriconazole susceptibility *in vitro* and during infection

We next wanted to test whether ZfpA affects antifungal susceptibility during infection. Triazoles are a first-line therapy for invasive aspergillosis that suppress fungal growth by impairing ergosterol synthesis and membrane integrity [35]. Further, *zfpA* was upregulated in a previous study of the transcriptional response of *A. fumigatus* to voriconazole treatment [16]. To assess voriconazole susceptibility in the ZfpA mutants, we measured colony diameter after 4 days of growth on solid GMM supplemented with 0.1 or 0.25 µg/mL voriconazole. To account for the effects of ZfpA manipulation on hyphal development and colony size, we report changes in growth for each strain as colony diameter relative to growth on GMM. At both concentrations tested, ZfpA overexpression reduced voriconazole susceptibility relative to WT CEA10 (Figs 6A and S2); while the effect of ZfpA deletion was less pronounced with a slight decrease in Δ*zfpA* relative colony diameter compared to WT CEA10 at 0.25 µg/mL (Figs 5A and S2).

Antifungals can work in concert with the host to clear fungal infection, and therefore drug efficacy and the mechanisms driving fungal killing can vary between *in vitro* and *in vivo* scenarios [25]. Our lab has successfully used voriconazole to treat *A. fumigatus* infection in larval zebrafish and previously reported that voriconazole completely protects larvae from death at 1 µg/mL [25]. Therefore, we selected a sub-effective dose of 0.1 µg/mL to screen for differences in voriconazole susceptibility during infection. We injected neutrophil-deficient Rac2D57N

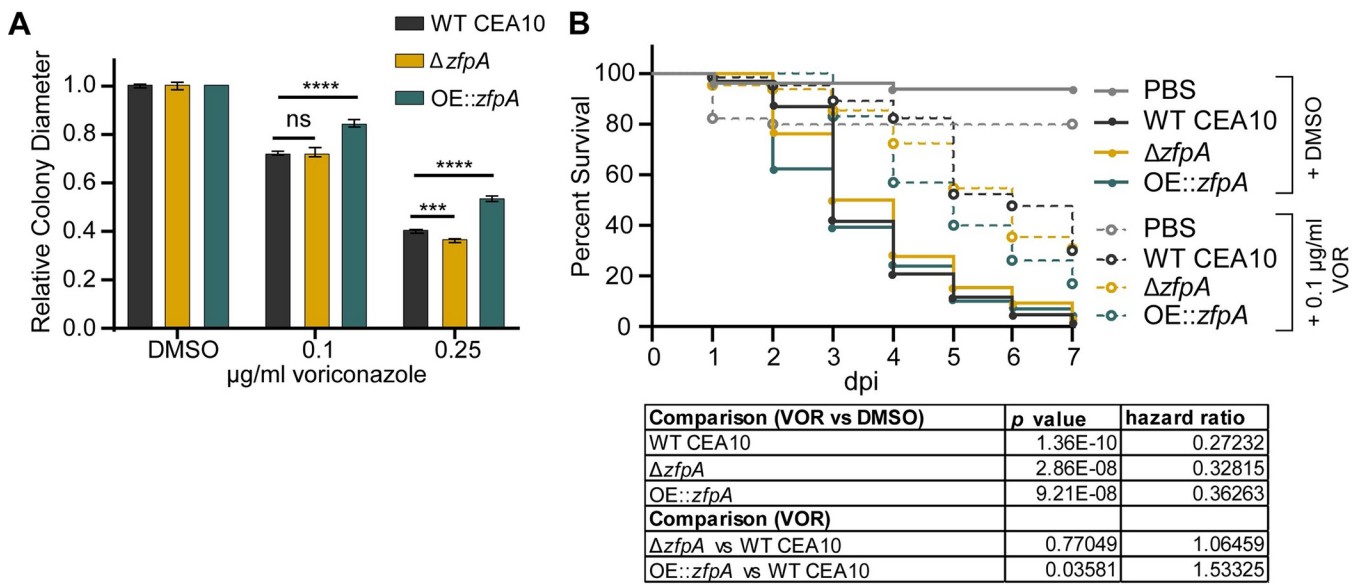

**Fig 6. ZfpA overexpression decreases voriconazole efficacy *in vitro* and during infection.** (A) Susceptibility of WT CEA10, ΔzfpA, and OE::zfpA to 0.1 and 0.25 μg/ml voriconazole (VOR). $10^4$ spores were point-inoculated on solid GMM with voriconazole or DMSO. Bars represent mean±s.d. of relative colony diameter of 4 plates per condition. *p* values calculated by ANOVA with Tukey's multiple comparisons. ***$p$<0.001, ****$p$<0.0001. (B) Survival analysis of infected Rac2D57N larvae (neutrophil-deficient) bathed in 0.1 μg/ml voriconazole or 0.001% DMSO. Average spore dose injected: WT CEA10 = 39, ΔzfpA = 35, OE::zfpA = 37. Results represent pooled data from 3 independent experiments. n = 44–66 larvae per condition. *p* values and hazard ratios calculated by Cox proportional hazard regression analysis.

larvae with spores of WT CEA10, ΔzfpA, or OE::zfpA, and added 0.1 μg/mL voriconazole to the larval water. As expected, voriconazole treatment improved survival of all larvae relative to the solvent-treated controls. However, voriconazole was least effective in animals infected with OE::zfpA (Fig 6B). Loss of ZfpA did not improve efficacy of voriconazole when compared to WT CEA10, similar to observations in our *in vitro* analyses (Fig 6A).

## ZfpA is required for echinocandin tolerance

We have previously shown that ZfpA deletion increases susceptibility to cell wall perturbations and the echinocandin caspofungin (Fig 5) [11]. Further, *zfpA* transcript levels increase in response to caspofungin exposure (S4 Fig) [15]. Here, we wanted to expand our analysis to include multiple echinocandins and test the effect of ZfpA overexpression on tolerance to this class of antifungals. To assess caspofungin susceptibility in the ZfpA mutants, we measured relative colony diameter after 4 days of growth on solid GMM supplemented with 0.25, 0.5, 1, and 8 μg/mL caspofungin or micafungin. As expected, ΔzfpA was most susceptible to caspofungin up to 1 μg/mL. ZfpA overexpression significantly improved caspofungin tolerance at these same concentrations (Figs 7A and S2, S5–S7 Movies). We selected 8 μg/mL to test whether ZfpA mutants were capable of paradoxical growth, a phenomenon in which drug efficacy decreases with increased drug concentrations [36]. At 8 μg/mL, colony diameter expanded for all strains, indicating that ZfpA is not essential for paradoxical growth. Micafungin exhibited greater inhibition of colony growth than caspofungin, with all strains having severely restricted growth at all concentrations tested (Figs 7B and S2 and S3). Similar to the effects of caspofungin, ΔzfpA was the most susceptible while OE::zfpA was most tolerant. There was no evidence of paradoxical growth at these concentrations of micafungin.

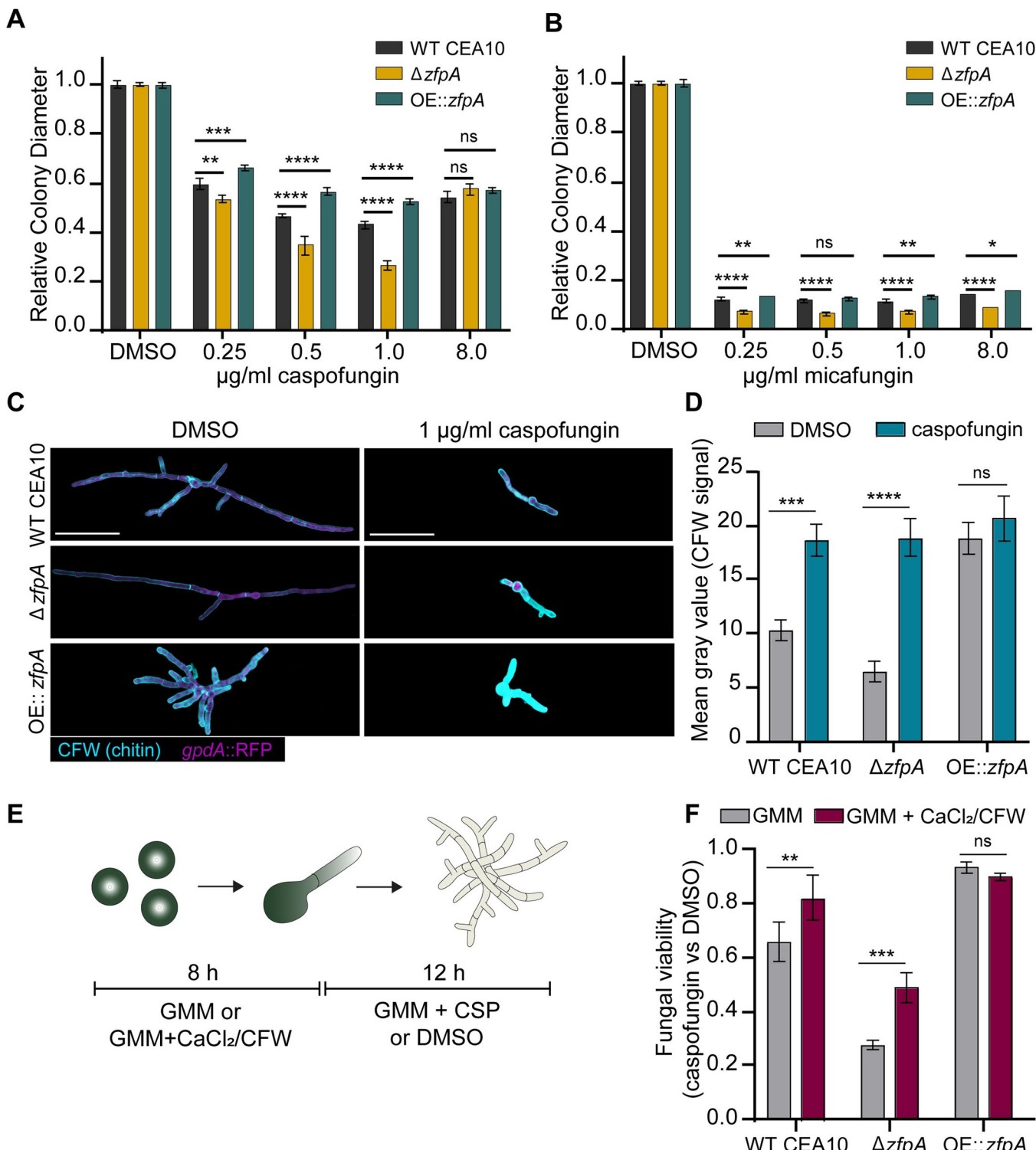

**Fig 7. ZfpA mediates echinocandin tolerance by altering developmental chitin synthesis.** (A) Susceptibility of WT CEA10, Δ*zfpA*, and OE::*zfpA* to 0.25, 0.5, 1, and 8 µg/ml caspofungin (CSP). $10^4$ spores were point-inoculated on solid GMM with caspofungin or DMSO. Bars represent mean±s.d. of colony diameter at 4 days post inoculation of 4 plates per condition. (B) Susceptibility of WT CEA10, Δ*zfpA*, and OE::*zfpA* to 0.25, 0.5, 1, and 8 µg/ml micafungin (MCF). $10^4$ spores were point-inoculated on solid GMM with micafungin or DMSO. Bars represent mean±s.d. of colony diameter at 4 days post inoculation of 4 plates per condition. *p* values calculated by ANOVA with Tukey's multiple comparisons. (C) Images represent calcofluor white (CFW) staining of WT CEA10, Δ*zfpA*, and OE::*zfpA* following overnight exposure to 1 µg/mL caspofungin (CSP) or DMSO. CFW staining is represented by cyan and cytoplasmic RFP signal is shown in magenta. Scale bar = 50 µm. (D) Mean gray value of CFW signal following DMSO or caspofungin treatment. Bars represent mean±s.e.m of 3

independent experiments. n = 24–30 hyphae per condition. (E) Experimental setup for chitin stimulation with CaCl$_2$/CFW. Spores were incubated for 8 h at 37°C or until germination in liquid GMM or liquid GMM supplemented with 0.2 M CaCl$_2$ and 100 µg/mL CFW. After germination, media was replaced for GMM + 1 µg/mL caspofungin or DMSO and hyphae were incubated for an additional 12 h before detecting PrestoBlue viability reagent signal in a plate reader. (F) Bars represent mean±s.d. of relative fungal viability following caspofungin exposure. Relative viability was calculated by normalizing the mean signal of caspofungin-treated wells to the mean signal of DMSO-treated wells. All experiments included 5 wells/condition. Data are pooled from 3 independent experiments. *p* values calculated by ANOVA with Sidak's multiple comparisons. *$p < 0.05$, **$p < 0.01$, ***$p < 0.001$, ****$p < 0.0001$.

## ZfpA-mediated changes in basal chitin content underpin differences in caspofungin tolerance

Caspofungin exposure stimulates a compensatory increase in chitin synthesis that is associated with decreased drug susceptibility [37]. We have previously reported that ZfpA deletion decreases chitin, while overexpression drastically increases chitin in the cell wall [11]. To test whether ZfpA is involved in compensatory chitin synthesis in response to caspofungin, we grew WT CEA10, Δ*zfpA*, and OE::*zfpA* in liquid GMM supplemented with 1 µg/mL caspofungin and visualized chitin content with calcofluor white (CFW) staining and fluorescence microscopy. As seen previously [11], Δ*zfpA* had reduced chitin and OE::*zfpA* had increased chitin relative to WT CEA10 (Fig 7C and 7D). Notably, all strains increased chitin in response to caspofungin, suggesting that ZfpA is not required for compensatory chitin production during caspofungin exposure (Fig 7C and 7D).

Despite the ability of Δ*zfpA* to upregulate chitin during drug exposure, it still displayed increased susceptibility to caspofungin compared to WT and OE::*zfpA* (Fig 6A and 6B). We thus hypothesized that temporal control of chitin synthesis is important for caspofungin tolerance. To test this hypothesis, we increased cell wall chitin prior to caspofungin exposure by pretreating spores with a combination of CaCl$_2$ and CFW to activate the Ca$^{2+}$-calcineurin and PKC (protein kinase C) stress response pathways responsible for maintenance of cell wall integrity [37]. Spores were grown in GMM or GMM supplemented with CaCl$_2$/CFW for 8 hours before exchanging media for GMM with or without caspofungin (Fig 7E). We measured fungal viability after 12 hours of caspofungin exposure using PrestoBlue viability reagent, which relies on the reducing environment of live cells to convert resazurin to fluorescent resorufin. CaCl$_2$ and CFW pretreatment improved viability of WT CEA10 by 16% and by 21% in Δ*zfpA* compared to untreated controls (Fig 7F). However, Δ*zfpA* viability was still lower than WT CEA10. There was no effect on OE::*zfpA*, which maintained high tolerance to caspofungin with and without pretreatment (Fig 7F). These data suggest that ZfpA-mediated changes in basal chitin levels during hyphal development are largely responsible for the differences in caspofungin tolerance among ZfpA mutants.

## Loss of ZfpA enhances caspofungin susceptibility during infection

As ZfpA is a determinant of caspofungin tolerance *in vitro*, we wanted to test the importance of ZfpA for caspofungin tolerance in a live host. We injected Rac2D57N larvae with WT CEA10, Δ*zfpA*, or OE::*zfpA* spores and added 1 µg/mL caspofungin to the larval water. Caspofungin had only a slight protective effect in larvae infected with WT CEA10 relative to the solvent-treated controls (Fig 8), in agreement with the reported fungistatic nature of caspofungin. No protective effect was seen in larvae infected with OE::*zfpA* (Fig 8), contrary to our observations during voriconazole treatment (Fig 6B). Survival of Δ*zfpA*-infected animals was significantly improved relative to controls (Fig 8), consistent with our *in vitro* analyses (Figs 7A, 7F and S2). This enhanced survival of Δ*zfpA*-infected animals was in contrast to voriconazole treatment where infections were indistinguishable between Δ*zfpA* and WT CEA10 (Fig 6B), suggesting that ZfpA may regulate features of *A. fumigatus* development specifically important for echinocandin susceptibility.

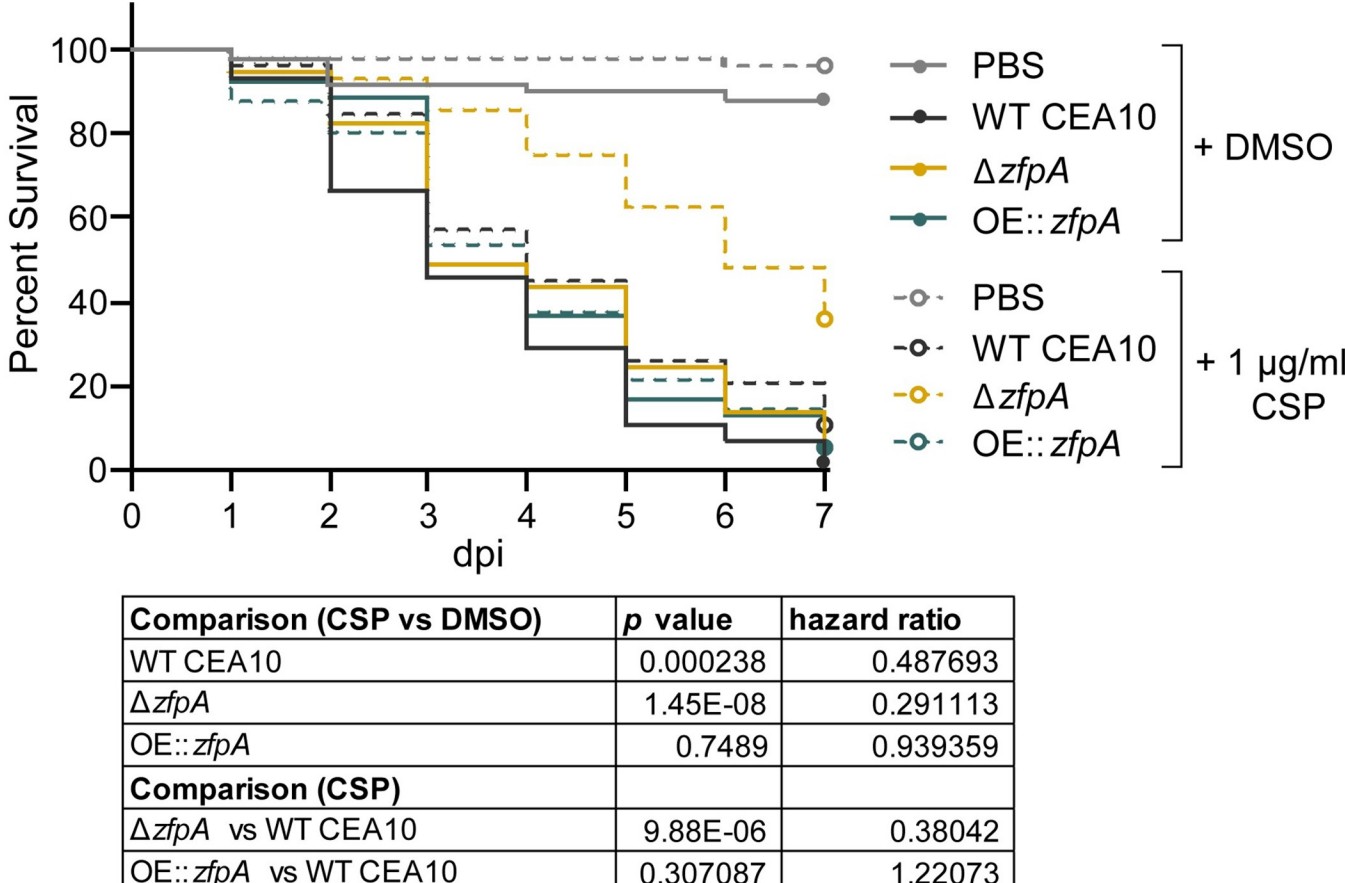

**Fig 8. ZfpA alters susceptibility to caspofungin during infection.** Survival analysis of Rac2D57N larvae (neutrophil-deficient) infected with WT CEA10, Δ*zfpA*, or OE::*zfpA* strains and bathed in 1 μg/mL caspofungin (CSP) or 0.01% DMSO. Average spore dose injected: WT CEA10 = 35, Δ*zfpA* = 30, OE::*zfpA* = 40. Results represent pooled data from 3 independent experiments. n = 50–59 larvae per condition. *p* values and hazard ratios calculated by Cox proportional hazard regression analysis.

## Discussion

Tissue invasive hyphae are a characteristic feature of IA pathogenesis permitted by failure of the host immune response to contain fungal growth, and effective IA treatment relies on hyphal clearance by antifungal drugs. However, the mechanisms that regulate hyphal growth during infection are not fully understood. This study identifies the transcription factor ZfpA as a regulator of hyphal resistance to immune cell- and antifungal-induced stress during *A. fumigatus* infection.

ZfpA deletion attenuated virulence and decreased fungal burden in immunocompetent hosts but did not affect virulence in neutrophil-deficient animals. Neutrophil-deficiency, or neutropenia, is not the only predisposing condition for *Aspergillus* infection. Infections occur with other immunosuppressive conditions such as long-term corticosteroid use or inborn errors in immunity [1]. Our findings suggest that ZfpA activity may be especially relevant in hosts with some preserved neutrophil function.

ZfpA deletion increased, while overexpression decreased, susceptibility to neutrophils but not to reactive oxygen species both *in vitro* and in PHOX-deficient zebrafish larvae, suggesting that ZfpA is important for resistance against non-oxidative killing mechanisms. Alterations in cell wall composition have been previously shown to impact virulence and neutrophil killing

of *A. fumigatus*. For example, the exopolysaccharide galactosaminogalactan (GAG) is a virulence factor that specifically mediates resistance to neutrophil extracellular traps [31]. Although we have not assessed GAG levels of the ZfpA mutants, we know ZfpA deletion decreases, while overexpression increases, chitin deposition. Genetic depletion of *A. fumigatus* chitin synthases or pharmacologic inhibition of chitin synthesis has been previously shown to increase susceptibility to neutrophil killing *in vitro* and attenuate virulence in corneal infection of mice [38]. It is unclear whether chitin protects against neutrophils by serving as a physical barrier to antimicrobial effectors or if it impacts phagocyte recognition of other immunologically relevant cell wall polysaccharides like β-1,3-glucan. ZfpA deletion and overexpression resulted in no detectable changes to neutrophil recruitment, IL-8 release, NETosis, or degranulation suggesting that potential changes in immunomodulatory cell wall components have minimal impact on neutrophil activation. Given the similar response of neutrophils to the ZfpA mutants, we suspect that the susceptibility of Δ*zfpA* is due to decreased hyphal structural integrity. However, more comprehensive studies comparing cell wall composition of the ZfpA mutants will be needed to fully appreciate how ZfpA mediates hyphal-phagocyte interactions.

How does hyphal branching impact virulence and interactions with neutrophils? Previous *in vitro* analyses of neutrophil-hyphae interactions suggest that branching is a double-edged sword. It serves as an evasive maneuver to escape neutrophils but may also increase opportunities for neutrophils to exert their microbicidal functions [14]. Septum formation is closely associated with branching and creates physical barriers within hyphae to protect hyphal compartments from damage. During infection of mice, septation deficiency severely limits tissue invasion and virulence, however, it is unclear whether these phenotypes result from decreased hyphal strength or ability to withstand host immunity [39]. While the pleiotropic effects of ZfpA manipulation make it challenging to determine the precise contributions of septation and branching to virulence of these strains, the live-imaging techniques used in this study provide some insights on how ZfpA protects against the host immune response. Repeated live-imaging of infected larvae revealed successful colonization of host tissue by the ZfpA deletion mutant. This experiment suggests the branching and septation defects of this strain do not limit tissue invasion but may contribute to the decreased fungal burden observed later in infection. We speculate this is due to increased susceptibility to phagocytes, as we saw this strain was completely unable to escape neutrophil killing *in vitro*. The hyperbranching ZfpA overexpression strain did not escape surrounding neutrophils more frequently than wild-type *A. fumigatus* yet survived longer, suggesting that increased branching was not advantageous in this *in vitro* scenario. We suspect that any potential detrimental effects of excessive branch production in this strain are offset by enhanced cell wall integrity and stress resistance.

The ZfpA-mediated changes to *A. fumigatus* hyphae that impact resistance to host defenses are likely also responsible for our observations of altered antifungal drug susceptibility. Our results indicate that ZfpA is more important for resistance to caspofungin and micafungin than to voriconazole, similar to previous analyses of a ZfpA deletion mutant [18]. In all experiments, ZfpA overexpression decreased susceptibility to echinocandins and voriconazole, both *in vitro* and during infection. While ZfpA deletion consistently increased susceptibility to echinocandins, changes in voriconazole susceptibility were dose dependent. Septa contribute to hyphal survival during echinocandin [39,40] and azole exposure [31], however, the necessity for septation has only been demonstrated with echinocandins [40]. Therefore, it is possible that increased septation induced by ZfpA overexpression provides protection against both classes of antifungals, while decreased septation in the deletion mutant has a more significant impact on echinocandin survival.

Upregulation of chitin synthesis during echinocandin exposure is considered a canonical adaptive stress response to β-1,3-glucan depletion that limits drug efficacy by restoring cell

wall integrity [36]. Moreover, increased chitin content is associated with increased caspofungin tolerance [37,41]. Thus, our observation that the chitin-depleted ZfpA deletion and chitin-rich overexpression strains have altered caspofungin susceptibility is not surprising. However, our finding that compensatory chitin synthesis is maintained in the ZfpA deletion mutant suggests that ZfpA may not have an essential role in restoring cell wall integrity during caspofungin exposure. Caspofungin and other cell wall stressors activate the cell wall integrity (CWI) pathway, a series of stress response signals that converge on a mitogen activated protein kinase (MAPK) cascade to activate transcriptional regulators of *de novo* cell wall synthesis [42]. Treating *A. fumigatus* spores with $CaCl_2$ and CFW stimulates chitin synthesis through activation of two pathways involved in CWI, the $Ca^{2+}$-calcineurin and PKC (protein kinase C) stress response pathways, respectively [37,41]. Given that $CaCl_2$/CFW pretreatment effectively increased caspofungin tolerance in the ZfpA deletion mutant and that *zfpA* expression is downstream of cAMP-PKC signaling during voriconazole exposure [16], we suspect the CWI pathway remains intact with ZfpA deletion. Collectively, our data suggest ZfpA regulation of chitin synthesis during hyphal development, and possibly septation, underlie changes in caspofungin tolerance.

Here, we describe a protective role for the transcription factor ZfpA in defense against two essential fungicidal effectors, the host immune response and antifungal treatment. Our findings provide new insights on genetic regulation of hyphal stress tolerance during infection. Future characterization of the ZfpA regulatory program should provide valuable targets for enhancing *A. fumigatus* susceptibility to both phagocytic and antifungal assault.

## Materials and methods

### Ethics statement

Animal care and use protocol M005405-A02 was approved by the University of Wisconsin-Madison College of Agricultural and Life Sciences (CALS) Animal Care and Use Committee. This protocol adheres to the federal Health Research Extension Act and the Public Health Service Policy on the Humane Care and Use of Laboratory Animals, overseen by the National Institutes of Health (NIH) Office of Laboratory Animal Welfare (OLAW).

### Fish lines and maintenance

Adult zebrafish and larvae were maintained as described previously [19]. Larvae were anesthetized in E3 water (E3) + 0.2 mg/mL Tricaine (ethyl 3-aminobenzoate, Sigma) prior to all experiments. To prevent pigment formation, larvae used in live-imaging experiments were treated with E3 + 0.2 mM N-phenylthiourea (PTU, Sigma) beginning at 1 day post fertilization (dpf). All zebrafish lines used in this study are listed in Table 1.

**Table 1. Zebrafish lines used in this study.**

| Line | Description | Reference |
|---|---|---|
| WT (AB) | ZL1 | ZIRC |
| Tg(*mpx:mCherry-2A-rac2WT*) | wildtype Rac2 and mCherry expressed in neutrophils—wildtype control strain for rac2D57N | [27] |
| Tg(*mpx:mCherry-2A-rac2D57N*) | Rac2D57N and mCherry expressed in neutrophils | [27] |
| Tg(*lyz:BFP/mpeg1:GFP*) | BFP-expressing neutrophils, GFP-expressing macrophages | [49] |
| p22*phox sa11798* | Point mutation in *cyba* (p22-encoding gene) leads to premature stop and loss of p22 protein | [22]; Zebrafish International Resource Center (ZIRC) |

**Table 2.** *Aspergillus* strains used in this study.

| Parental Background | Strain | Genotype | Description | Reference |
|---|---|---|---|---|
| CEA10 | TCDN6.7 | *ΔakuB;; argB-; gpdA::RFP::argB; pyrG-* | RFP-expressing pyrG-<br>Used to generate TDGC1.2, TJW215.1, TJW216.1 | This study |
| | TDGC1.2 | *ΔakuB; argB-; gpdA::RFP::argB; pyrG-; fumipyrG* | RFP-expressing wildtype | This study |
| | TJW215.1 | *ΔakuB; gpdA::RFP::argB; argB-; pyrG-; ΔzfpA::parapyrG* | RFP-expressing ΔzfpA | This study |
| | TJW216.1 | *ΔakuB; argB-; gpdA::RFP::argB; pyrG-; parapyrG::gpdA(p)::zfpA* | RFP-expressing OE::zfpA | This study |
| Af293 | TFYL80.1 | *fumiargB; argB-; ΔnkuA::mluc; pyrG-* | ΔnkuA pyrG- | [50] |
| | TJW213.1 | *fumiargB;argB-;ΔnkuA::mluc; ΔzfpA::parapyrG; pyrG-;* | ΔzfpA | This study |
| | TJW214.1 | *fumiargB;argB-;ΔnkuA::mluc; parapyrG::gpdA(p)::zfpA; pyrG-;* | OE::zfpA | This study |
| | TFYL81.5 | *fumiargB;fumipyrG-;ΔnkuA::mluc;pyrG1 argB1* | Wildtype Af293 | [51] |

### *Aspergillus* strains and growth conditions

*Aspergillus fumigatus* conidial stocks were maintained at -80˚C in glycerol suspension until being streaked on solid Glucose minimal media (GMM), supplemented with the appropriate amounts of uridine (0.5 mg/mL), uracil (0.5 mg/mL), or arginine (1 mg/mL) when necessary. Liquid GMM with 0.5% yeast extract was used to extract genomic DNA. Conidia were harvested from solid GMM culture grown in darkness at 37˚C for 3–4 days for a short-term (1 month at 4˚C) working stock by scraping with an L-shaped cell spreader in 0.01% Tween-water. The conidial suspension was then passed through sterile miracloth to remove hyphal fragments. To prepare conidia for microinjection, $10^6$ conidia were plated on solid GMM and grown in darkness at 37˚C for 3–4 days. Conidia were harvested as described above and the conidial suspension was centrifuged at 900 *x g* for 10 minutes at room temperature. The resulting pellet was resuspended in 1X PBS and spun again. Conidia were resuspended in 1X PBS, passed through sterile miracloth, and counted using a hemacytometer. The conidial concentration was adjusted to 1.5 x $10^8$ conidia/mL for the injection stock (1 month at 4˚C). All *Aspergillus* strains used in this study are listed in Table 2. A step-by-step protocol for *Aspergillus* infection of larval zebrafish is provided in Schoen et al., 2021 [43].

### Generation of ZfpA deletion and overexpression strains

For *zfpA* deletion strains, two 1 kb DNA fragments immediately upstream and downstream of *zfpA* open reading frame (ORF), were amplified by PCR from Af293 genomic DNA, and were fused to 2 kb *A. parasiticus pyrG* fragment from pJW24 [44] using double joint PCR. Fungal transformation to TCDN6.7 was performed following a previously described approach [45]. Transformants were confirmed for targeted replacement of the native locus by PCR and Southern blotting using *Pci*I restriction enzyme digests and both the P-32 labeled 5′ and 3′ flanks (S2 Fig) to create TJW215.1 from TCDN6.7. For *zfpA* overexpression strains, two 1 kb fragments immediately upstream and downstream of *zfpA* translational start site were amplified by PCR from Af293 genomic DNA. *A. parasiticus pyrG::A. nidulans gpdA(p)* were used as the selectable marker and overexpression promoter, respectively, and were amplified from the plasmid pJMP9 [46]. The three fragments were fused by double joint PCR and transformed into TCDN6.7 to create strain TJW216.1. Single integration of the transformation construct was confirmed by PCR and Southern blotting using *Pci*I restriction enzyme digests and both the P-32 labeled 5′ and 3′ flanks (S3 Fig). To create the prototrophic wildtype control strain TDGC1.2 from TCDN6.7, 2 kb *A. fumigatus pyrG* was amplified to complement *pyrG* auxotrophy. All of primers for this study is listed in Table 3. DNA extraction, restriction enzyme digestion, gel electrophoresis, blotting, hybridization, and probe preparation were performed by standard methods [47].

**Table 3. Primers used in this study.**

| Name | 5' -> 3' | Use |
|---|---|---|
| zfpA5'F | TGACCATGATCTCCACTTCCCC | *zfpA* deletion |
| zfpA5'R | CGATATCAAGCTATCGATACCTCGACTCGC AGACGTCCTAAGCTCGATAGTCGACTG | *zfpA* deletion |
| parapyrGF | GAGTCGAGGTATCGATAGCTTG | *zfpA* deletion |
| parapyrGR | ATTCGACAATCGGAGAGGCTGC | *zfpA* deletion |
| zfpA3'F: | GTCGCTGCAGCCTCTCCGATTGTCGAATCG ACGATGAACCTGAGGAAGATGACGACG | *zfpA* deletion |
| zfpA3'R: | GATACTTTTCAGCTGCAGCCGC | *zfpA* deletion |
| zfpAkoconfF: | CACAGCGCATAAAACCATCGCC | Confirmation of *zfpA* deletion |
| zfpAkoconfR: | TAGGGCCTATCCTTAGGGTACC | Confirmation of *zfpA* deletion |
| zfpAOE5'F | zfp5'F recycle | *zfpA* overexpression |
| zfpOE5'R | CCAATTCGCCCTATAGTGAGTCGTATTACG GCAGACGTCCTAAGCTCGATAGTCGACTG | *zfpA* overexpression |
| OEPyGF: | CGTAATACGACTCACTATAGGGC | *zfpA* overexpression |
| OEPyGR: | GGTGATGTCTGCTCAAGCGGG | *zfpA* overexpression |
| zfpOE3'F: | CAGCTACCCCGCTTGAGCAGACATCACCAT GCAGAGCCCAGGAGAACATTCCGAC | *zfpA* overexpression |
| zfpOE3'R: | GTATTCGCACGTAACGATGGGG | *zfpA* overexpression |
| zfpAOEconfF | ATTCATCTTCCCATCCAAGAACC | Confirmation of *zfpA* overexpression |
| zfpOEconfR | TGTTTGCTCAACGCCATGCACG | Confirmation of *zfpA* overexpression |
| afumipyrGF | CTACCTCGAGAATATGCCTCAAAC | pyrG- complementation |
| afumipyrGR | GGCGACTTATTCTGTCTGAGAG | pyrG- complementation |

## *In vitro* chemical perturbation assays

To assess radial growth, GMM plates supplemented with 0.25, 0.5, 1, and 8 µg/mL caspofungin or micafungin, or 0.1 or 0.25 µg/mL voriconazole were point inoculated with $10^4$ spores for each strain and grown at 37°C for four days before measuring colony diameter. To assess cell wall, osmotic, and ROS stress tolerance, square plates were inoculated with $10^5$, $10^4$, $10^3$, and $10^2$ spores in a volume of 2 µL of each strain and grown at 37°C for 48 hours. All plates were solid GMM supplemented with 30 µg/mL CFW, 1.2 M sorbitol, or 3 mM $H_2O_2$. Both radial growth and dilution plating experiments were completed in triplicate or quadruplicate.

## Spore microinjections

Larvae (2 dpf) were anesthetized and 3 nL of *A. fumigatus* conidial suspension was microinjected into the hindbrain ventricle via the otic vesicle as previously described [19,43]. The conidial stock was mixed 2:1 with 1% Phenol Red prior to injection to visualize the inoculum in the hindbrain. After injection larvae were rinsed 3X with E3 without methylene blue (E3-MB) to rinse off Tricaine and remained in E3-MB throughout all experiments. Larvae were transferred to individual wells of a 96-well plate for survival experiments or individual wells of 24- or 48-well plates for imaging experiments. For survival experiments, larvae were checked daily for 7 days and considered dead if there was no visible heartbeat. To determine the number of conidia injected for each experiment, 8 larvae/condition were collected after injection and individually added to microcentrifuge tubes in 90 µL 1X PBS with 500 µg/mL kanamycin and 500 µg/mL gentamycin. Larvae were homogenized for 15 sec using a mini-bead beater and then plated on solid GMM plates. Colony forming units (CFUs) were counted and averaged after 2–3 days incubation at 37°C. The CFU averages for each condition and experiment are reported in figure legends.

## Zebrafish drug treatments

Caspofungin (Cat# 501012729, Fisher) and voriconazole (PZ0005, Sigma) were reconstituted in DMSO at 1 mg/mL and stocks were stored in small aliquots at -20°C to avoid repeated freeze-thaw cycles. Larvae were treated with caspofungin diluted 1:1,000 (f.c. 1 μg/mL) in E3-MB and the media was exchanged daily for fresh drug solution. Larvae were treated with voriconazole diluted 1:10,000 (f.c. 0.1 μg/mL) in E3-MB and the media was exchanged daily for fresh drug solution.

## Calcofluor white (CFW) staining and caspofungin treatment

To visualize chitin content, 2,500 spores were grown in 1 mL GMM with 0.1% DMSO or 1 μg/mL caspofungin for 14 h on a glass coverslip in individual wells of a 12-well plate. Coverslips were then rinsed once with PBS and inverted on a 200 μL drop of 0.1 mg/mL CFW for 10 min. Coverslips were then washed with water for 10 min on a rocker and mounted on slides immediately before imaging. CFW was kept at room temperature in darkness at a stock concentration of 1 mg/mL in water.

## CaCl$_2$/CFW treatment and PrestoBlue viability assay

For CaCl$_2$ and CFW treatment, 2.5 x 10$^5$ spores in 100 μL liquid GMM or GMM supplemented with 0.2 M CaCl$_2$ and 0.1 mg/mL CFW were plated in a 96-well plate and incubated at 37°C for 8 h. Media was then removed from the germlings and replaced with GMM + 0.1% DMSO or 1 μg/mL caspofungin and incubated for 11 h at 37°C. After 11 h, media was replaced with GMM + PrestoBlue viability reagent (f.c. 1:10, ThermoFisher Cat# P50200) with 0.1% DMSO or 1 μg/mL caspofungin. Plates were then incubated at 37°C for an additional hour before fluorescence was measured at 555/590 nm using a PerkinElmer Victor3V plate reader.

## Human neutrophil isolation and co-incubation with *A. fumigatus*

All blood samples were obtained from healthy donors and were drawn according to the University of Wisconsin-Madison Minimal Risk Research Institutional Review Board-approved protocol (ID: 2017–0032) per the Declaration of Helsinki. Formal written consent was obtained from donors prior to blood draw. Neutrophils were isolated immediately after blood collection using the MACSxpress Whole Blood Neutrophil Isolation Kit (Miltyeni Biotec #130-104-434) and manufacturer instructions. Neutrophils were centrifuged for 5 min at 200 *x g* and the pellet was resuspended in 1 mL PBS for counting. Neutrophils were centrifuged again and resuspended to a final concentration of 4 x 10$^5$ cells/mL in RPMI + 2% fetal bovine serum and used immediately. For live-imaging of neutrophil-fungal interactions, 2 x 10$^3$ spores/well were grown until the germling stage (8 h at 37°C) in 500 μL liquid GMM in a 24-well plate. GMM was then removed and replaced with 500 μL of the neutrophil suspension (200,000 neutrophils, neutrophil, spore 100:1). The 24-well plate was then immediately brought to the microscope for imaging.

## SYTOX Green staining

Wells of black 96-well plates were coated with 30 μg/mL bovine fibronectin (Sigma) overnight at 4°C prior to experiments. Spores were grown in the fibronectin-coated 96-well plates for 8 h at 37°C in GMM until the germling stage (100,000 spores/well in 100 μl). Following germination, GMM was removed and freshly isolated primary human neutrophils were added to wells at a concentration of 200,000 cells/well in 100 μl RPMI + 2% FBS. The plate was then incubated at 37°C for 30 min to let the cells rest prior to stimulating positive controls wells (neutrophils

only) with 50 μl of 100 μg/ml PMA (Phorbol myristate acetate, f.c. 100 nM, Sigma). Wells with unstimulated neutrophils were used as a negative control. The plate was incubated for an additional 6 h at 37˚C until adding 50 μl SYOX Green to all wells (f.c. 375 nM, ThermoFisher). Fluorescence (485nm/535nm, 0.1s) was measured 15 min after addition of SYTOX Green using a PerkinElmer Victor3V plate reader.

## IL-8 and Myeloperoxidase detection

Wells of 96-well plates were coated with 30 μg/mL bovine fibronectin (Sigma) overnight at 4˚C prior to experiments. Spores were grown in the fibronectin-coated 96-well plates for 8 h at 37˚C in GMM until the germling stage (100,000 spores/well in 100 μl). Following germination, GMM was removed and freshly isolated primary human neutrophils were added to wells at a concentration of 200,000 cells/well in 100 μl RPMI + 2% FBS. For every experiment 5 wells were used for each condition. Wells with unstimulated neutrophils were used as a negative control. Supernatants were collected after 3 h of co-incubation and centrifuged for 3 min at 300 $x g$ to pellet any remaining cells. The supernatant was then removed and stored at -80˚C. IL-8 was measured using the Human IL-8/CXCL8 DuoSet ELISA kit (R&D Systems) following manufacturer protocol. Supernatants were diluted 1:4 to fit within the standard curve. Myeloperoxidase was measured using the Human Myeloperoxidase DuoSet ELISA kit (R&D Systems) following manufacturer protocol. Supernatants were diluted 1:1000 to fit within the standard curve.

## Image acquisition

Transgenic larvae were pre-screened for fluorescence using a zoomscope (EMS3/SyCoP3; Zeiss; Plan-NeoFluor Z objective). For multi-day imaging experiments, larvae were anesthetized and mounted in a Z-wedgi device [43,48] where they were oriented such that the hindbrain was fully visible. Z-series images (5 μm slices) of the hindbrain were acquired on a spinning disk confocal microscope (CSU-X; Yokogawa) with a confocal scan head on a Zeiss Observer Z.1 inverted microscope, Plan-Apochromat NA 0.8/20x objective, and a Photometrics Evolve EMCCD camera. Between imaging sessions larvae were kept in E3-MB with PTU in individual wells of 24- or 48-well plates. Neutrophil-fungal interactions were imaged using an inverted epifluorescence microscope (Nikon Eclipse TE3000) with a Nikon Plan Fluor 20x/0.50 objective, motorized stage (Ludl Electronic Products) and Prime BSI Express camera (Teledyne Photometrics). Environmental controls were set to 37˚C with 5% $CO_2$. Images were acquired every 3 min for 12 h. Imaging of *A. fumigatus* stained with CFW was performed using an upright Zeiss Imager.Z2 LSM 800 laser scanning confocal microscope with Airyscan detection and a Plan-Apochromat 20x /0.8 objective. A single z plane image was acquired for each hypha. Images were captured using identical laser and exposure settings for each condition.

## Image analysis and processing

Images of larvae in Fig 2A represent maximum intensity projections of z-series images generated in FIJI. For analysis of germination, fungal burden, and immune cell recruitment, z-series images were converted to max intensity projections using FIJI. Germination was scored as the presence or absence of germinated conidia as defined by the presence of a germ tube. Fungal burden and immune cell recruitment were analyzed by manually thresholding the corresponding fluorescent channel and measuring the 2D area of the fluorescent signal in the infected region. No alterations were made to images prior to analysis. Brightness and contrast were adjusted in FIJI to improve definition and minimize background signal for presentation

purposes only. In Fig 4B, germlings were recorded as dead at the image frame in which the cytoplasmic RFP signal was no longer visible. Images in Fig 7C represent a single image acquired for each hypha. Brightness was adjusted equally for all presented images. For measurement of CFW signal, hyphae were outlined and the mean gray value of the hyphae were measured using FIJI.

## Statistical analyses

The number of independent replicates (*N*) and larvae, germlings, or plates (*n*) used for each experiment are reported in the figure legends. Survival analyses of larvae and fungal germlings were performed with RStudio using Cox proportional hazard regression analysis with experimental condition included as a group variable, as previously described [19]. Pair-wise *P* values and hazard ratios are included in the main figure or figure legend for all survival experiments. Analysis of germination rate and percent of germlings to escape neutrophils were performed with Student's *t*-tests (GraphPad Prism version 9). 2D area of fungal growth, neutrophils, and macrophages represent least-squared adjusted means±standard error of the mean (LSmeans ±s.e.m.) and were compared using ANOVA with Tukey's multiple comparisons (RStudio). Relative colony diameters in Figs 5, 6 and S3 were compared using ANOVA with Tukey's multiple comparisons (GraphPad Prism version 9). Comparison of fungal viability in Fig 7F was done using ANOVA with Sidak's multiple comparisons (GraphPad Prism version 9). All graphical representations of data were created in GraphPad Prism version 9 and figures were ultimately assembled using Adobe Illustrator (Adobe version 23.0.6).

## Supporting information

**S1 Fig. Survival of wild-type and neutrophil-deficient larvae infected with ZfpA mutants in Af293 background.** Survival analysis of larvae with the dominant negative Rac2D57N neutrophil mutation (neutrophil-deficient) or wild-type siblings injected with PBS, WT Af293, Δ*zfpA*, or OE::*zfpA* strains. WT larvae average spore dose injected: WT Af293 = 27, Δ*zfpA* = 29, OE::*zfpA* = 25. Rac2D57N larvae average spore dose injected: WT Af293 = 38, Δ*zfpA* = 32, OE::*zfpA* = 35. Results represent pooled data from 2 independent replicates. n = 46–48 larvae per condition. *p* values and hazard ratios calculated by Cox proportional hazard regression analysis.
(TIF)

**S2 Fig. Radial growth of ZfpA mutants exposed to voriconazole, caspofungin, and micafungin.** (A) Susceptibility of WT CEA10, Δ*zfpA*, and OE::*zfpA* to 0.1 and 0.25 μg/mL voriconazole (VOR). $10^4$ spores were point-inoculated on solid GMM with voriconazole or DMSO. Images of voriconazole plates are representative of colony growth 4 days post inoculation. (B) Susceptibility of WT CEA10, Δ*zfpA*, and OE::*zfpA* to 0.25, 0.5, 1, and 8 μg/mL caspofungin (CSP). $10^4$ spores were point-inoculated on solid GMM with caspofungin or DMSO. Images of caspofungin plates are representative of colony growth 5 days post inoculation. (C) Susceptibility of WT CEA10, Δ*zfpA*, and OE::*zfpA* to 0.25, 0.5, 1, and 8 μg/mL micafungin (MCF). $10^4$ spores were point-inoculated on solid GMM with micafungin or DMSO. Images of micafungin plates are representative of colony growth 4 days post inoculation.
(TIF)

**S3 Fig. ZfpA mediates micafungin tolerance.** Susceptibility of WT CEA10, Δ*zfpA*, and OE::*zfpA* to 0.05, 0.125, 0.25, and 1 μg/mL micafungin. $10^4$ spores were point-inoculated on solid GMM with micafungin or DMSO. Bars represent mean±s.d. of colony diameter at 4 days post inoculation of 4 plates per condition. *p* values calculated by ANOVA with Tukey's multiple

comparisons. *$p<0.05$, **$p<0.01$, ****$p<0.0001$.
(TIF)

**S4 Fig. Expression of *zfpA* during caspofungin exposure.** Bars represent fold change in *zfpA* expression in Af293 and CEA10 backgrounds during exposure to 0.2 and 2 μg/ml caspofungin. Fold change values were collected from RNAseq dataset in supplementary materials of Colabardini et al., 2022 [15].
(TIF)

**S5 Fig. Southern confirmation of *ΔzfpA* mutants.** Genomic DNA was digested by *Pci*I. Wild-type (8.1 kb for Af283, 8.3 kb for CEA10), and *ΔzfpA* (5.3 and 3.4 kb for Af293, 5.5 and 3.4 kb for CEA10). TJW213.1 and TJW215.1 were chosen for subsequent experiments.
(TIF)

**S6 Fig. Southern confirmation of *OE::zfpA* mutants.** Genomic DNA was digested by *Pci*I. Wildtype (8.1 for Af293, 8.3 kb for CEA10), and *OE::zfpA* (6.1 and 5.5 kb for Af293, 6.3 and 5.5 kb for CEA10). TJW214.1 and TJW216.1 were chosen for subsequent experiments.
(TIF)

**S1 Movie. Interactions between neutrophils and wild-type CEA10 germlings.** Representative movie of neutrophils engaging with two WT CEA10 germlings. One germling loses cytoplasmic RFP signal and is killed while the other escapes surrounding neutrophils. Images were acquired every 3 min for 12 h. Left panel: brightfield. Right panel: *A. fumigatus* cytoplasmic RFP. Scale bar = 20 μm. 10 frames/s.
(M4V)

**S2 Movie. Interactions between neutrophils and Δ*zfpA* germling.** Representative movie of neutrophils engaging with Δ*zfpA* germling. The germling does not escape surrounding neutrophils and loses cytoplasmic RFP signal within 30 min of co-incubation. Images were acquired every 3 min for 12 h. Left panel: brightfield. Right panel: *A. fumigatus* cytoplasmic RFP. Scale bar = 20 μm. 10 frames/s.
(M4V)

**S3 Movie. Interactions between neutrophils and OE::*zfpA* germlings.** Representative movie of neutrophils engaging with two OE::*zfpA* germlings. One germling does not escape surrounding neutrophils and loses cytoplasmic RFP signal after 225 min of co-incubation while the other escapes. Images were acquired every 3 min for 12 h. Left panel: brightfield. Right panel: *A. fumigatus* cytoplasmic RFP. Scale bar = 20 μm. 10 frames/s.
(M4V)

**S4 Movie. Neutrophils exhibit swarming behavior in response to *A. fumigatus* germling.** Example of primary human neutrophils swarming around *A. fumigatus* germling. Germling is indicated by black arrow. The first neutrophil contact is indicated by a blue asterisk. Note the morphology change of surrounding neutrophils after this first cell makes contact and the subsequent rapid accumulation of neutrophils around the germling. Scale bar = 20 μm. 2 frames/s.
(M4V)

**S5 Movie. WT CEA10 growth and lysis during caspofungin exposure.** Representative movie of WT CEA10 germling exposed to 1 μg/ml caspofungin. Note loss of RFP signal with cell lysis. Images acquired every 5 min. Scale bar = 50 μm. 10 frames/s.
(MP4)

**S6 Movie. Δ*zfpA* growth and lysis during caspofungin exposure.** Representative movie of Δ*zfpA* germling exposed to 1 μg/ml caspofungin. Note loss of RFP signal with cell lysis. Images acquired every 5 min. Scale bar = 50 μm. 10 frames/s.
(MP4)

**S7 Movie. OE::*zfpA* growth during caspofungin exposure.** Representative movie of OE::*zfpA* germling exposed to 1 μg/ml caspofungin. Note that hyphae do not lyse. Images acquired every 5 min. Scale bar = 50 μm. 10 frames/s.
(MP4)

## Acknowledgments

We thank members of the Huttenlocher and Keller labs for helpful discussions of the research and manuscript.

## Author Contributions

**Conceptualization:** Taylor J. Schoen, Anna Huttenlocher, Nancy P. Keller.

**Formal analysis:** Taylor J. Schoen, Morgan A. Giese.

**Funding acquisition:** Anna Huttenlocher, Nancy P. Keller.

**Investigation:** Taylor J. Schoen, Dante G. Calise, Morgan A. Giese.

**Methodology:** Taylor J. Schoen.

**Resources:** Dante G. Calise, Jin Woo Bok, Chibueze D. Nwagwu, Nancy P. Keller.

**Supervision:** Anna Huttenlocher, Nancy P. Keller.

**Visualization:** Taylor J. Schoen.

**Writing – original draft:** Taylor J. Schoen.

**Writing – review & editing:** Dante G. Calise, Jin Woo Bok, Morgan A. Giese, Chibueze D. Nwagwu, Robert Zarnowski, David Andes, Anna Huttenlocher, Nancy P. Keller.

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
