## [Decision Letter · Decision Letter 0]

27 Feb 2023

Dear Dr. Keller,

Thank you very much for submitting your manuscript "Aspergillus fumigatus transcription factor ZfpA regulates hyphal development and alters susceptibility to antifungals and neutrophil killing during infection" for consideration at PLOS Pathogens. As with all papers reviewed by the journal, your manuscript was reviewed by members of the editorial board and by several independent reviewers. In light of the reviews (below this email), we would like to invite the resubmission of a significantly-revised version that takes into account the reviewers' comments.

While the reviewers thought that the work was interesting, there was a consistent concern that the mechanism by which the zfpA mutant was more susceptible to neutrophil killing was not adequately explored. Please pay special attention to the comments of Reviewers #1 and #2.

We cannot make any decision about publication until we have seen the revised manuscript and your response to the reviewers' comments. Your revised manuscript is also likely to be sent to reviewers for further evaluation.

Sincerely,

Scott G. Filler, M.D.

Guest Editor

PLOS Pathogens

Alex Andrianopoulos

Section Editor

PLOS Pathogens

Kasturi Haldar

Editor-in-Chief

PLOS Pathogens

orcid.org/0000-0001-5065-158X

Michael Malim

Editor-in-Chief

PLOS Pathogens

orcid.org/0000-0002-7699-2064

Reviewer's Responses to Questions

**Part I - Summary**

Reviewer #1: In this study, the authors characterize the role of the Aspergillus fumigatus transcription factor ZfpA in resistance to neutrophil attack and antifungal treatment. They utilize a zebrafish infection model to show that this transcription factor is crucial in virulence because it protects hyphae against attack by neutrophils. Further, infections in neutrophil-defective zebrafish demonstrate that the zfpA mutant is exquisitely hypersensitive to the echinocandin caspofungin during infection, in contrast to a lack of hypersensitivity to voriconazole. This is a rigorous study reporting potentially important new connections among fungal cell wall structure, antifungal resistance and host-pathogen interaction. The use of the zebrafish model provides important context for the activity of ZfpA during infection and therapeutic drug treatment.

Reviewer #2: This manuscript by Schoen et al examines the outcomes of infection with a ZfpA mutant in a zebrafish model. The authors nicely show that in WT zebrafish the ZfpA is less virulent and this correlates with enhanced susceptibility to PMN antifungal killing activity. However, the authors fail to link what changes the susceptibility of the ZfpA mutant to PMN killing and what antifungal effector mechanisms from the PMN cells are utilized. Rather the authors go on to explore the cell wall integrity pathway and susceptibility to the antifungal drugs particularly chitin synthase inhibitors. They demonstrate that ZfpA mutant is more susceptible to caspofungin.

Reviewer #3: Keller et al Plos Pathogens:

The authors recently reported that the transcription factor ZfpA has an important role in hyphal development. In the current study, these investigators use a ΔzfpA mutant to demonstrate the importance of ZfpA in virulence and in resistance to immunity in a zebrafish model of infection and in response to human neutrophils. Further, the use of well characterized anti-fungal agents identify a specific role for ZfpA in chitin synthesis. Overall, the data as presented are clear and in general support the conclusions. However, there are a few points that the authors should address.

**Part II – Major Issues: Key Experiments Required for Acceptance**

Reviewer #1: None.

Reviewer #2: Experimentally the authors nicely demonstrated that WT zebrafish infected with the ZfpA are less susceptible to infection (Figure 1A) and this correlates with enhanced susceptibility to PMN antifungal killing activity (Figure 3B/C).

1. Why is the ZfpA mutant more susceptible to PMN killing? Do the cell wall alterations change the ability of the fungal germlings to induced PMN activation?

2. What antifungal effector mechanisms from the PMN cells are utilized to control the ZfpA mutant? Are phox-deficient PMNs unable to kill the ZfpA mutant germlings or is an alternative mechanism (PMN degranulation or NETosis)?

Reviewer #3: Fig 2,3. Quantification of the area with hyphae is certainly one method of showing total hyphae. However, although they state that loss of RFP implies killing, unless I am mistaken, the presence of RFP does not address viability. Can the authors use CFU or another method in addition imaging, which would strengthen their conclusions? Also, in addition to showing percent RFP, the authors should also show RFP values.

Fig 4. The authors state that there were no visible differences between the colonies of Aspergillus strains grown in sorbitol or hydrogen peroxide. However, the examples they give appear to have differenced in the colonies of the OE::zfpA (less filamentation on the edges) when compared with WT or ΔzfpA strains at the 103 and 102 inoculums. The authors should show quantitative data based on repeat experiments.

Fig 5, 6. In contrast to Figure 4, here they state that there are differences in colony diameter. However, in the examples shown, this differences are not apparent. Overall, the agar plate results shown in Fig5A and Fig6A,C are unconvincing.

Fig 7. The authors claim that the susceptibility of the ΔzfpA is due to reduced baseline chitin levels. While microscopy in Fig7A supports this conclusion, they need to quantify CFW intensity.

**Part III – Minor Issues: Editorial and Data Presentation Modifications**

Reviewer #1: Minor issues:

1) It is not clear whether the role of ZfpA is direct or indirect in regulation of chitin levels

2) It is not directly demonstrated if increased chitin is the specific causative factor in increased echinocandin resistance. Does ZfpA control other aspects of physiology that promote resistance/tolerance to echinocandins?

3) Please spend more space describing previous work on ZfpA and explain how this study improves upon that previous work—relative to susceptibility to antifungals and immune attack (Ref 15-18)

4) It would be interesting to see if the results with neutrophil-defective fish are the same or different from corticosteroid-treated fish, to examine different types of immunocompromise that are associated with IA. This seems like a model that has been used before in zebrafish

5) It would be useful to examine b-glucan and chitin exposure in the zfpA and OEX-ZFPA strains, as well as immune cytokine responses

6) The % of germlings alive was quantified in experiments with human neutrophils in vitro, providing a good measure for resistance to immune attack. How much does survival of zfpA germlings increase in neutrophil-defective fish? Also, it seems like another relevant measure would be the rate of hyphal segment death. Is it possible to measure this in time-lapse or time-course experiments in fish?

7) It would be useful to have a more nuanced discussion of the idea forwarded in the Discussion that these results suggest combining chitin inhibition and septation inhibition would be an effective drug combination. Since chitin inhibition with CFW (in the context of high calcium) increases ECH resistance, combinations of cell wall inhibitors may not always yield the desired synergies—due to inherent compensatory programs.

8) Please explain what PrestoBlue is measuring, and comment on the correlation between PrestoBlue activity and loss/retention of cytoplasmic fluorescence

9) Please speculate on your expectations for the relative activity of ZfpA in control of these phenotypes in other isolates of Af, such as Af293

10) Does CFW/CaCl2 treatment alter ZpfA expression or activity?

11) In line 194, it seems like the nomenclature delta-zfpA would be more appropriate (typo)

Reviewer #2: (No Response)

Reviewer #3: none

PLOS authors have the option to publish the peer review history of their article (what does this mean?). If published, this will include your full peer review and any attached files.

Reviewer #1: No

Reviewer #2: No

Reviewer #3: No
---

## [Editor Report · Decision Letter 1]

18 Apr 2023

Dear Dr. Keller,

We are pleased to inform you that your manuscript 'Aspergillus fumigatus transcription factor ZfpA regulates hyphal development and alters susceptibility to antifungals and neutrophil killing during infection' has been provisionally accepted for publication in PLOS Pathogens.

Best regards,

Scott G. Filler, M.D.

Guest Editor

PLOS Pathogens

Alex Andrianopoulos

Section Editor

PLOS Pathogens

Kasturi Haldar

Editor-in-Chief

PLOS Pathogens

orcid.org/0000-0001-5065-158X

Michael Malim

Editor-in-Chief

PLOS Pathogens

orcid.org/0000-0002-7699-2064
---

## [Editor Report · Acceptance letter]

26 Apr 2023

Dear Dr. Keller,

We are delighted to inform you that your manuscript, "*Aspergillus fumigatus* transcription factor ZfpA regulates hyphal development and alters susceptibility to antifungals and neutrophil killing during infection," has been formally accepted for publication in PLOS Pathogens.

Best regards,

Kasturi Haldar

Editor-in-Chief

PLOS Pathogens

orcid.org/0000-0001-5065-158X

Michael Malim

Editor-in-Chief

PLOS Pathogens

orcid.org/0000-0002-7699-2064